# Ge Ion Implanted Photonic Devices and Annealing for Emerging Applications

**DOI:** 10.3390/mi13020291

**Published:** 2022-02-12

**Authors:** Xingshi Yu, Xia Chen, Milan M. Milosevic, Weihong Shen, Rob Topley, Bigeng Chen, Xingzhao Yan, Wei Cao, David J. Thomson, Shinichi Saito, Anna C. Peacock, Otto L. Muskens, Graham T. Reed

**Affiliations:** 1Optoelectronics Research Centre, University of Southampton, Southampton SO17 1BJ, UK; xy1m13@soton.ac.uk (X.Y.); xia.chen@soton.ac.uk (X.C.); M.Milosevic@soton.ac.uk (M.M.M.); shenweihong@sjtu.edu.cn (W.S.); rob.topley@gmail.com (R.T.); Xingzhao.Yan@soton.ac.uk (X.Y.); wei.cao@soton.ac.uk (W.C.); d.thomson@soton.ac.uk (D.J.T.); acp@orc.soton.ac.uk (A.C.P.); o.muskens@soton.ac.uk (O.L.M.); 2State Key Laboratory of Advanced Optical Communication Systems and Networks, Shanghai Jiao Tong University, Shanghai 200240, China; 3Zhejiang Lab, Hangzhou 311100, China; chenbg@zhejianglab.com; 4Electronics and Computer Science, University of Southampton, Southampton SO17 1BJ, UK; ss1a11@ecs.soton.ac.uk

**Keywords:** silicon photonics, optical waveguide, Ge ion implantation, annealing, wafer-scale testing, post-fabrication trimming, programmable photonic circuits

## Abstract

Germanium (Ge) ion implantation into silicon waveguides will induce lattice defects in the silicon, which can eventually change the crystal silicon into amorphous silicon and increase the refractive index from 3.48 to 3.96. A subsequent annealing process, either by using an external laser or integrated thermal heaters can partially or completely remove those lattice defects and gradually change the amorphous silicon back into the crystalline form and, therefore, reduce the material’s refractive index. Utilising this change in optical properties, we successfully demonstrated various erasable photonic devices. Those devices can be used to implement a flexible and commercially viable wafer-scale testing method for a silicon photonics fabrication line, which is a key technology to reduce the cost and increase the yield in production. In addition, Ge ion implantation and annealing are also demonstrated to enable post-fabrication trimming of ring resonators and Mach–Zehnder interferometers and to implement nonvolatile programmable photonic circuits.

## 1. Introduction

Silicon photonics is currently a commercially established and yet fast-growing technology for communication systems. It has also started to play a significant role in many other applications, including chemical and biomedical sensing, LiDAR application in driving assistance systems, and in applications for building quantum networks. The integration with III–V light source materials such as indium phosphide through bonding has also been reported to solve the lack of intrinsic light emission within the silicon platform [1,2,3,4,5,6,7,8,9].

To monitor the status of fabrication processes and characterise the performance of photonic circuits after production, wafer-scale testing is an essential technology in a semiconductor production line [10,11]. However, testing the silicon photonics devices optically at wafer scales is nontrivial. Various techniques can be used to couple the optical signal from optical fibres to photonic chips, such as prism coupling, butt coupling, end-fire coupling, and grating coupling [12,13,14,15]. Among them, grating couplers provide an effective coupling technique that can couple between integrated waveguides and the out-of-plane fibres at the wafer scale without dicing the photonic chips [16,17]. However, grating couplers are normally only used as the input/output for the whole photonic circuit. Therefore, apart from measuring the performance of the entire circuit during wafer-scale testing, it is not possible to test any subsystems or individual devices on the photonic chips. Once a problem within the circuits, such as a fabrication error, occurs, early identification and analysis can bring huge cost savings. Therefore, it is desirable to routinely test some key sensitive subsystems or individual devices on the photonic chips, at a wafer scale. Erasable grating couplers and directional couplers are, therefore, proposed for such applications, based on germanium (Ge) ion implantation and annealing technologies [18,19].

With current semiconductor fabrication technology, the uniformity of the fabrication processes for some silicon photonics devices, such as ring resonators and Mach–Zehnder interferometers (MZIs), exhibits variability. The optical phase error induced by the variation in waveguide dimensions across the chip will significantly change the device performance. This is one of the major factors affecting the yield of final silicon photonic modules in production [20,21]. For example, the resonant wavelength and Q factor of ring resonators will be affected by variations in the fabrication process [22,23]. The transmission of MZI mainly depends on the phase difference between the two arms, and this will also suffer from the variations in the fabrication process [24,25].

Silicon photonic modulators and filters have benefited from the resonance properties of ring resonators. However, this structure is extremely susceptible to fabrication variations [26]. Several methods have been investigated to tune the resonant wavelength of optical ring resonators. Electron beam and laser irradiation are two techniques that have been used for trimming, with each utilising a different process to realise a refractive index change [27]. Inducing strain and compaction into an oxide cladding by e-beam can change the effective index of the optical mode in the devices [28,29]. The polymethyl methacrylate cladding allows the resonant wavelength of ring resonators to be trimmed after irradiation by ultraviolet light [30]. Post-fabrication trimming can also be realised by introducing photosensitive materials onto the waveguide surface and subsequently exposing them to lasers [31]. The optical properties of the waveguide can also be shifted by depositing and partial etching of a thin SiN film on top of devices [32]. Laser-induced lattice damage can also change the refractive index of silicon waveguides and can provide a shift in resonant wavelength [33]. In this study, we demonstrated that Ge ion implantation and annealing can be applied for post-fabrication trimming [34,35,36,37], with some unique advantages over other techniques, such as a large change in refractive index change and easy implementation.

In addition to the applications of wafer-scale testing and post-fabrication device trimming, we also implemented a proof of principle of nonvolatile programmable photonic circuits with the Ge implantation technology [38]. Based on the refractive index change associated with Ge ion implantation and annealing mentioned above, the flow of an optical signal within the photonic circuit can be switched or rerouted using erasable directional couplers or trimmable MZIs. Such devices were used as the basic building blocks for large-scale programmable photonics circuits. After fine-tuning the transmission, the working states of the directional couplers or MZIs are fixed and do not need a continuous electrical power supply to retain the operating point [34]. This will greatly reduce the power consumption, compared with more traditional programmable photonic circuits, implemented with MZI arrays controlled by integrated thermal heaters [39].

## 2. Ion Implantation and Annealing

### 2.1. Ge Ion Implantation

Ion implantation is a widely used fabrication process to achieve impurity doping in a solid target [40]. Typically, in the CMOS industry, ions of elements from groups III and V are implanted to change the free carrier concentration. The process to alter the charge density present in the sample via doping is different from the ion implantation process used in this study to change the refractive index of the silicon waveguide via crystal damage. For crystalline silicon waveguides, lattice disorder can be induced during the implantation process. Therefore, the effective index of propagating mode in optical devices can be altered through this process via a damage-induced change in the refractive index. During the implantation process, the ions are accelerated in an electric field and injected into the sample, causing lattice damage in the crystalline silicon. The alteration in the refractive index is attributed directly to this lattice damage [41,42]. Therefore, a variety of implanted species can be utilised to cause lattice damage.

In this project, Ge was chosen as the implanted ion in the ion implantation process to create amorphous sections in a crystalline silicon waveguide. Firstly, Ge is a group IV element, which means that the free carrier concentration of the silicon waveguide will not be changed. Secondly, it is a CMOS-compatible element, and compared with carbon and silicon, Ge has a higher relative mass and can cause greater lattice damage than other group IV elements for the same implanted dose [41,43]. Consequently, amorphisation can be achieved for a lower dose than for lighter ions. Ion implantation is also a temperature-dependent process. Low ion implantation temperature can reduce the required ion implantation dose and implant time [42]. However, it also increases the utilisation time of the implantation equipment and production costs because self-heating via the ion beam must be avoided. The lattice disorder in silicon crystal has been reported with a very mild temperature dependency below 323 K [44]. Therefore, room temperature can be used to reduce fabrication costs.

Figure 1 shows the lattice damage profile in a 220 nm thick silicon waveguide [45]. The King and King 3D software packages were used to simulate the damage profile of silicon waveguides after Ge ion implantation [46,47,48]. The energy and implantation dose in this simulation are 100 keV and 1 × 10^15^ ions/cm^2^, respectively. The real part of the refractive index of the implanted silicon waveguide regions was increased by approximately 0.5 due to 80% lattice disorder or more, which was taken to be the definition of amorphisation for the purposes of this research. The induced lattice defects in silicon will also result in a small increase (0.04) of the imaginary part of the refractive index [49].

The mask patterns to define the impanated area (i.e., implantation masks) were fabricated using a DUV lithography scanner. Since shrinkage of a photoresist mask layer during the ion implantation can occur, a SiO_2_ layer was used as a hard mask, to eliminate this uncertainty in the fabrication process. The ion implantation process was carried out at the Ion Beam Centre at the University of Surrey. An ion energy value of 130 keV and a dose of 1 × 10^15^ ions/cm^2^ were used in order to efficiently implant Ge into silicon and create deeper implantation in the silicon waveguides, based on our previous studies [50].

### 2.2. Annealing

The lattice damage in silicon, created by implantation in the waveguide, can be repaired by a thermal heating process. The refractive index will, therefore, decrease during this recrystallisation process [51]. Several annealing methods were used in this study—namely, rapid thermal annealing (RTA), laser annealing, and heating via an integrated electrical heater.

Rapid thermal annealing (RTA) is a widely used process in standard CMOS fabrication lines. It can heat the whole silicon wafers/chips, in order to improve the electrical properties, and it activates dopants, changes the properties of films, and anneals the implantation damage. The temperature of the annealing process can be precisely controlled [35,52]. Therefore, it is a useful tool to characterise refractive index variations at specific temperatures but is not useful for local device heating.

In our study, continuous wave-laser exposure and pulse-laser exposure were used to achieve the annealing of implanted sections in waveguides. More details of laser annealing setups can be found in our previously published studies [34,35,36,37,38]. Compared with RTA, this method can locally anneal the implanted regions of single devices. As the energy of the annealing laser photons is directly transferred into the lattice of the sample, the annealing process is accomplished by scanning the laser spot over the implanted regions, a process that can be completed in a few seconds [53,54]. Although laser annealing is quicker than RTA, it is still not an ideal annealing method in large-scale production. Achieving the correct power density on the sample surface can be time-consuming, and the devices cannot be processed by laser irradiation after packaging.

Integrated electrical heaters have been used in photonic devices to correct the operating points and realign the resonant wavelength of ring resonators via the thermo-optic effect [55]. However, we can also achieve annealing via integrated microheaters on top of optical devices. The target area on the chip can be heated to the annealing temperature by applying the correct voltage to the heater. This approach can enable localised and rapid annealing, and facilitate multiple simultaneous annealing points in large-scale photonic integrated circuits, even after packaging.

Results from our previous study revealed that the difference in residual insertion loss between the crystalline silicon and annealed silicon devices is negligible. The resultant peak concentration of Ge ions in the silicon waveguides is less than 0.3%, compared with a silicon concentration in the same volume [18,34,45,50], indicating that the creation of a SiGe alloy is not a concern.

## 3. Erasable Grating Couplers

A typical structure of a conventional grating coupler is based on periodically etched waveguides to cause a periodic variation in the effective refractive index. By means of the Ge ion implantation, erasable grating couplers can be fabricated using the lattice damage induced by Ge ion implantation in silicon waveguide to cause a similar periodic change in the index, thus diffracting the light and maintaining the surface planarity at the same time. After annealing, the implanted grating coupler can be totally erased, become a transparent waveguide and make no difference to the entire circuit, which are significant advantages for wafer-scale testing.

For proof of principle of implanted gratings for wafer-scale testing, a bow-tie-shaped design was fabricated, as shown in Figure 2. In the implanted grating, the etched part of a conventional grating was replaced by the amorphous regions induced by Ge ion implantation. After sweeping the parameter values of grating period and implantation depth at a fixed duty cycle of 0.5, we chose a period of 600 nm and 130 nm implanted depth, to obtain an optimal coupling efficiency of 44%. The resulting implanted depth is almost twice the depth of the etched grating depth. This is because the lower Δn of the amorphous to the crystalline interface, compared with that of silicon-to-air or -dioxide interface, results in a lower coupling strength for gratings of similar dimensions. Therefore, to achieve similar coupling efficiency to etched gratings, implanted gratings require deeper amorphous regions, to compensate for the smaller index contrast.

An implanted grating coupler such as this can be positioned anywhere within the optical circuit to evaluate the performance of individual components or to monitor any device or subsystem. Furthermore, after testing, it can be erased by annealing to eliminate any additional insertion loss introduced to the circuit.

A continuous-wave (CW) laser system was initially used for annealing [18]. To guarantee the regrowth of crystal lattice in silicon, the laser spot was scanned over the pattern of the implanted gratings. After full exposure, the implanted grating coupler can be completely erased. Material quality before and after annealing was measured by Raman spectroscopy, to determine whether amorphous regions were successfully annealed and recrystallised [18].

The coupling efficiency of the erasable grating coupler before and after annealing was measured. The implanted grating coupler shows a peak coupling efficiency of ~5.5 dB per coupler [45]. After laser annealing, light passes through the erased grating coupler with almost no additional loss and couples out via the conventional output grating coupler. The residual propagation loss through the erased grating coupler is suppressed to under −25 dB.

## 4. Implanted Mach–Zehnder Interferometer

The Mach–Zehnder interferometer (MZI) is one of the most popular components in photonics design. The working performance of a single MZI is controlled by the phase difference between the two arms. In the conventional design of an MZI, the phase difference can be induced by a length difference of the two arms and the thermo-optic effect, applied via an integrated heater. In our design, the variation of the phase difference is achieved by Ge ion implantation and subsequent annealing techniques, to choose a specific operating point.

Figure 3 illustrates a basic structure of an implanted MZI. The device was formed of rib waveguides with a 100 nm slab layer. Implanted sections were induced into each arm of the MZI with different lengths of 7 μm and 2 μm. The longer one was primarily responsible for the phase difference and, therefore, controlled the power distribution at the output ports. The shorter one was used to balance any unwanted reflection at the interface of the crystalline Si and implanted Si regions.

### 4.1. Post-Fabrication Trimming of MZI

For optical modulators, a specific operating point is usually desirable, such as the quadrature point. Due to the fabrication tolerances and environmental factors, the operating point of any MZI normally will be shifted, compared with the original design. Post-fabrication trimming of MZIs has been studied by several groups, and several effective methods were reported. A similar method was proposed by Samarao et al. [56], but a refractive index change of only 0.02 was achieved. In our study, Ge ion implantation can bring much higher refractive index changes, up to one order of magnitude larger.

To experimentally demonstrate this, a CW laser spot, with a diameter of 2.5 μm and power of 45 mW, was used to scan across the implanted sections in the two arms of an MZI (Figure 3). The measurement results of its transmission during the annealing process are shown in Figure 4.

Before annealing, the 5 μm difference of the implanted sections in both arms results in approximately 87% of the signal coupled into the output port 2. After 1 μm of the implanted section was annealed by a CW laser, over 95% of the signal was detected in port 2. Annealing the remaining 4 μm length can switch the output from port 2 to port 1, demonstrating full control of the operating point of the MZI.

### 4.2. Real-Time Phase Tuning of MZI

To implement trimming in commercial applications, real-time measurements are necessary. Therefore, we studied real-time trimming as proof of principle. The implanted MZI was annealed with a real-time optical measurement setup, which is shown in Figure 5. Optical signal T (mV) was collected by the photodetector, coupled to output 1. In this setup, a lock-in amplifier was used to extract the optical signal even within a high-noise environment.

During the annealing process (with reference to Figure 5), the annealing laser spot, with a power of 2.4 mW, was scanned linearly from the right-hand side of the 7 μm long implanted section to its left. As the phase difference between the two arms changes with the annealing process, the resultant transmission shift in the implanted MZI output is shown in Figure 6a. Taking the right-hand side of 7 μm long implanted section as the reference position (0 μm), the change in T starts from position 0 μm to 6.6 μm. Figure 6b shows the curve of ΔT, which is collected by the lock-in amplifier. ΔT is the differential of transmission T. In Figure 6b, ΔT starts at −1 mV. Then, this curve follows a downward trend and reaches the lowest point of the curve at 3 μm, which indicates the location of the quadrature point with the largest derivative of the MZI output. There is a small discontinuity of the ΔT curve, which is caused by the shaking of the piezo stage during the scanning. The phase-dependent transmission of the signal T should follow the trend of (1 + cosθ) where θ represents the phase difference between the two arms of MZI. Therefore, the derivative of this transmission should follow the trend of −sinθ. The negative maximum value of this curve is located at approximately −1.8 mV, corresponding to a phase difference of 0.5π between the arms. Based on the former discussion, it can be estimated that the whole phase shift corresponds to approximately 0.72π, after the pump laser annealing.

The average power of the annealing laser is 2.4 mW, which is insufficient to achieve full annealing for the implanted sections in one annealing cycle. Figure 6c,d show the phase-dependent shift, using θ as a free parameter, of T and ΔT in four annealing cycles. This result also shows consistency with former results [34].

### 4.3. Electrical Annealing of MZI

Electrical annealing of implanted MZIs with integrated TiN heaters was also demonstrated. The top and cross-sectional views of an implanted MZI are shown in Figure 7. A 600 nm thick SiO_2_ layer was deposited to separate the optical mode from the heater. The top SiO_2_ cladding layer was deposited to protect the heater filament from the air at high temperatures.

The optical transmission spectrum of an implanted MZI is shown in Figure 8. Over 90% of the optical signal is coupled into the drop port before annealing. We applied 7 V in this experiment, corresponding to electrical thermal power of 150 mW. After 10 s annealing time, approximately 90% of the input optical signal was detected at the through port of the MZI. Due to the phase error between the two arms of MZI, 10% insertion loss can be further improved by precisely controlling the annealing temperature and annealing length. After 10 s annealing time, the implanted sections were fully annealed. No further change in optical transmission was observed. The simulated temperature distribution (cross-section) of this device is shown in Figure 9. The temperature of the implanted waveguides was predicted to be around 600 °C to 700 °C in the annealing process.

## 5. Implanted Ring Resonators

Ring resonators represent the basic building component of many photonics circuits. They are widely used for optical switching, modulation, sensing, and optical signal filtering applications. However, these devices are also very susceptible to fabrication tolerances and environmental conditions; even a small variation in waveguide width or ambient temperature may result in a significant resonant wavelength shift; therefore, thermal heaters are typically required which will add to the overall power consumption of the photonic integrated circuit. The resonant wavelength shift (of the order of nanometres) caused by dimension variations are orders of magnitude larger than the shift (order of picometers) caused by ambient temperature variations [57]. Therefore, by utilising ion implantation technology and localised annealing, it is possible to induce localised refractive index changes and, therefore, trim the extinction ratio and resonant wavelength of fabricated optical filters. This process saves a significant amount of power required for accurate repositioning of the resonant wavelength due to a shift caused by fabrication tolerances.

Figure 10 shows a schematic of a Ge ion implanted silicon ring resonator. We used a 220 nm silicon-on-insulator (SOI) platform, to fabricate 500 nm wide rib waveguides (100 nm thick silicon slab layer; 120 nm etch depth) and ring resonators with different bend radii. By implanting a 6 µm long section of germanium into the silicon ring resonators of 10 µm in radius, we were able to locally increase the effective refractive index of silicon and, therefore, shift the resonant wavelength to a longer wavelength. We then performed localised laser annealing using the same setup used for trimming MZIs, to accurately position the resonant wavelength at any wavelength within the full free spectral range (FSR) of the ring resonator. Example transmission spectra before and after annealing are shown in Figure 11.

Ring resonator trimming was also performed in real time, and as can be seen from Figure 12, we were able to manually achieve a precision of 3% within the targeted wavelength position of 1550 nm for a variety of ring resonators, even when manually controlling the trimming time.

Although ion implantation and localised laser annealing represent very accurate procedures for trimming resonant wavelengths and do not require any additional fabrication steps for electrical heaters fabrication, electrical annealing is popular and is expected to become mainstream in silicon photonics. With silicon photonics applied to more industry applications, electrical heater annealing is more achievable in photonic circuits with high integration levels.

For racetrack resonators with 25 µm radii, Figure 13 shows the shift of the resonance peak around 1550 nm during the electrical annealing process. The applied voltage is 1.6 V, and the electrical power is approximately 43 mW, which can bring a wavelength shift of 0.5 nm. Figure 14 shows the wider shift of resonant wavelength under higher electrical powers, (59 mW, 120 mW, and 160 mW). Compared with laser annealing, the applied power and annealing time can be directly controlled, to adjust the resonant wavelength with high accuracy, resulting in a simpler operational process.

## 6. Erasable Directional Couplers

### 6.1. Erasable Directional Couplers

Directional couplers (DCs) have been reported as the basic building blocks for configurable photonic circuits. Compared with other devices that can also realise similar switching functions, such as MZIs and ring resonators, they have smaller dimensions. In the semiconductor and integrated photonics industries, the footprint has been an essential consideration in production cost. Furthermore, the performance of DCs is relatively insensitive to temperature fluctuations [58]. A cross-section of a typical device is shown in Figure 15a, and the simulated profile of lattice damage within the implanted waveguide is shown in Figure 15b. Implanted waveguides are successfully formed in the slab region of the conventional rib waveguide to couple light in/out, which can be erased by a laser annealing process.

In applications discussed so far in this paper, the optical signals propagate in traditional silicon waveguides fabricated by a dry etching process. However, in erasable DCs, the coupling waveguides were formed solely by an ion implantation process, which was first demonstrated in silicon photonics. Two types of DCs were designed and fabricated, as shown in Figure 15c (single-stage DC) and Figure 15d (two-stage DC). Optical microscope images of a typical fabricated single-stage DC and two-stage DC are shown in Figure 15e,f, respectively.

The devices were designed and optimised using the variational FDTD solver from Lumerical MODE solutions and fabricated in the cleanroom facilities in The University of Southampton, UK. The propagation loss of the germanium-ion-implanted waveguides with various widths was first characterised using the cut-back method. A loss of 32.6 dB/mm was measured for implanted waveguides with 560 nm width. Whilst this loss is very high, only very short lengths of a few micrometres are required in DCs. Figure 16a,b show the measured transmission of single-stage DCs and two-stage DCs, respectively. For single-stage DCs with a coupling length of around 6 μm, over 80% coupling efficiency was experimentally achieved. For two-stage DCs with a coupling length of around 12 μm, over 90% coupling efficiency was measured. The simulated results are also provided, shown as dotted lines in both figures.

### 6.2. One-Time Programmable Photonic Circuits

The previous devices were used to build 1 × 4 and 2 × 2 OTP (one-time programmable) switching circuits. No subsequent continuous electrical power supply is needed to maintain the operating point because the silicon lattice is permanently changed with our technology. Microscopic images of the switching circuits are shown in Figure 17.

The 1 × 4 switching circuit comprises one through port (P_4_) and three drop ports (P_1_, P_2_, and P_3_). These drop port waveguides are connected with the through port bus waveguide by three two-stage DCs in series. The input optical signal can be directed to any of these four outputs by suitable programming of the circuit. Less than −11.6 dB crosstalk is achieved between each channel.

Figure 17b shows a 2 × 2 OTP switching circuit comprising four two-stage DC structures. This 2 × 2 transmission array allows cross-coupling between two optical paths. Compared with the simple 2 × 2 switch, using two outputs and inputs of a single DC, our design offers lower crosstalk [59]. By using the annealing technique, the 2 × 2 OTP switching circuit can be permanently programmed into bar and cross-operating modes. According to the measurement results, crosstalk of −18.3 dB is achieved at 1550 nm for both channels.

### 6.3. Electrical Annealing of Directional Couplers

As laser annealing cannot usually be applied after the photonic devices are packaged, electrical annealing using integrated TiN heaters was also demonstrated for DCs. The structure of the implanted two-stage directional coupler with an integrated heater is shown in Figure 18. The design of the integrated heaters is similar to the ones used for trimming the MZIs introduced previously.

Figure 19 shows the transmission spectra of the implanted two-stage directional coupler before and after annealing. Before annealing, the directional coupler output exhibited a random level of approximately 65% of the optical signal at the drop port. The remaining 35% was detected at the through port. After electrical annealing with an applied voltage of 8 V, the signal at the drop port decreased to around 1.5%. Meanwhile, over 95% signal was detected at the through port. This indicates that most of the optical power remains in the input waveguide and is coupled out at the through port because the implanted waveguide was successfully annealed using the integrated heater.

## 7. Applications and Discussions

The ion implantation technique has been widely reported for modification and doping of materials. The performance of devices can, therefore, be improved after this treatment [60,61,62,63,64]. In previous experiments, Ge ion implantation was used to induce lattice damage into crystalline silicon waveguides, and the real part of the refractive index can be increased to approximately 3.96. The implantation process will not generate free carriers in silicon waveguides because, similar to silicon, Ge is a group IV material. The transmission of silicon optical devices can be shifted after the implantation process. This alteration of the performance of optical devices is larger than the typical variation brought about by fabrication errors, therefore allowing for correction or ‘trimming’ of such devices. The annealing process, also widely reported, is the mechanism utilised in removing the lattice defects created by implantation [65,66,67,68]. We used the annealing technique here to tune the performance of implanted optical devices for different applications, including wafer-scale testing, post-fabrication trimming, and programmable photonic circuits.

### 7.1. Wafer-Scale Testing

The traditional testing methods, such as butt coupling, are not suitable for large-scale comprehensive testing. They offer limited opportunities for accessing intermediate testing points in photonic circuits for monitoring individual optical devices within a photonic circuit. One possible solution is to use another device, such as an MZI or directional coupler, to tap a small portion of the optical signal from the testing point in a photonic circuit for monitoring and testing purposes. However, these couplers are not removable and will induce permanent optical loss throughout the operational life of photonic circuits [59,69].

On the other hand, our erasable grating couplers and directional couplers were demonstrated to enable flexible comprehensive wafer-scale testing for large-scale integrated photonic circuits without inducing additional loss during subsequent operation of the photonic circuits. The optical signal used for monitoring can be coupled in/out through implanted devices, which can be erased after testing. After the annealing process, the measured residual insertion loss is typically the order of −25 dB for the erasable grating couplers and −18 dB for the erasable directional couplers [18,19].

The implanted sections of these devices can be annealed by laser or electrical heaters, with the latter being a more feasible choice for commercial silicon photonic chips with high volume production and large-scale integration, as thermal heaters can be activated even in packaged devices. Tapers with a length of at least 700 μm are required for each erasable grating coupler, to minimise residual loss after annealing and to facilitate straightforward alignment within a commercial wafer prober, whereas erasable directional couplers can be much smaller [50]. A smaller insertion loss should be potentially achieved if a shorter implanted waveguide is used. Furthermore, there is typically no requirement to launch 90% of an optical signal for testing purposes.

### 7.2. Post-Fabrication Trimming of Optical Devices

Post-fabrication trimming techniques were investigated to fine-tune the optical phase in waveguide-based devices. In this study, the refractive index of the silicon crystalline waveguide was controlled by introducing implanted sections and annealing technologies. The effective index of a propagating mode and the working performance of devices, including MZIs, ring resonators, and directional couplers could, therefore, be trimmed.

For ring resonators, fabrication tolerances will influence the resonant wavelength, the accuracy of which is essential for applications such as sensing and modulation. In our study, the effective index change of the propagating mode in the ring waveguide depends on the length of the induced implantation section. This induced shift can also be controlled by annealing methods such as laser and integrated thermal heating. Our research also successfully demonstrated post-fabrication trimming of the critical coupling condition for racetrack resonators. To achieve this, the same technique was used to implant the slab layer between and bus waveguide and the ring waveguide. Then, the coupling efficiency could be trimmed through partial annealing of implanted section.

We also studied post-fabrication trimming of MZIs using similar techniques. Implanted sections were created in the two arms of MZIs. Then, an annealing process could gradually anneal the implanted waveguide sections and trim the operating point, hence the transmission performance. One example application is for optical modulation, where MZIs can offer the highest modulation efficiency working at the quadrature point.

### 7.3. Programmable Photonic Circuits

Based on the mature fabrication and integration technologies, photonic circuits can support complex applications. Programmable photonic circuits with application flexibility can be designed and developed to realise a variety of functions with the same chip when programmed in different ways. These circuits are built with waveguide meshes, phase shifters, and tuneable couplers, which can be programmed by software [39]. The flow of light in the programmable PICs can be electrically manipulated to realise different circuit functionality from the same chip. Conventional programmable circuits use an electrical signal to control integrated heaters to induce thermo-optic effects in thermal phase shifters. However, hundreds of heaters are typically required for such a system, which has also brought new issues into programmable photonics circuits, such as high power consumption, heat dissipation, and crosstalk between each phase shifter.

The erasable and tuneable optical devices investigated in this study can be used to fabricate nonvolatile actuators in programmable photonics circuits. By introducing a section of an implanted waveguide and then controlling annealing, the routing of an optical signal in an implanted MZI or directional coupler can be tuned. There are also other techniques proposed for building nonvolatile programmable circuits, such as phase-change materials [70,71] and mechanically latched MEMS [72,73]. However, our proposed programmable photonic circuits based on ion implantation and annealing techniques are more CMOS compatible.

## 8. Conclusions

We used Ge ion implantation and annealing technologies to realise erasable and trimmable silicon optical devices, including erasable grating couplers, erasable directional couplers, tuneable MZIs, and ring resonators. The applications of these devices were also discussed.

Erasable grating couplers and directional couplers were demonstrated as flexible testing points used for wafer-scale testing of photonic integrated circuits, which facilitates comprehensive circuit testing. These testing points can then be erased permanently after testing, with negligible residual insertion loss. Our proposed technology was also used in the post-fabrication trimming of silicon photonic devices with many unique advantages. For trimming of ring resonators, a large trimming range, up to 10 nm (across the entire free-spectral range) of 10 µm rings, was achieved. Table 1 illustrates a comparison of the current trimming methods of ring resonators.

For trimming of MZIs, a trimming accuracy of 0.078 rad and a trimming range of 1.2π were achieved with active feedback control. These trimmable and tuneable devices, such as MZIs and directional couplers, were also demonstrated for rerouting the optical signals of a photonic circuit after fabrication. This enabled us to design a multipurpose photonic integrated circuit with the capability to be programmed for various specific applications after fabrication or packaging. This one-time programmable photonic circuit can potentially reduce the overall production costs, with an increase in production volume, and can speed up the development or prototyping cycles for new photonic circuits.

## Figures and Tables

**Figure 1 micromachines-13-00291-f001:**
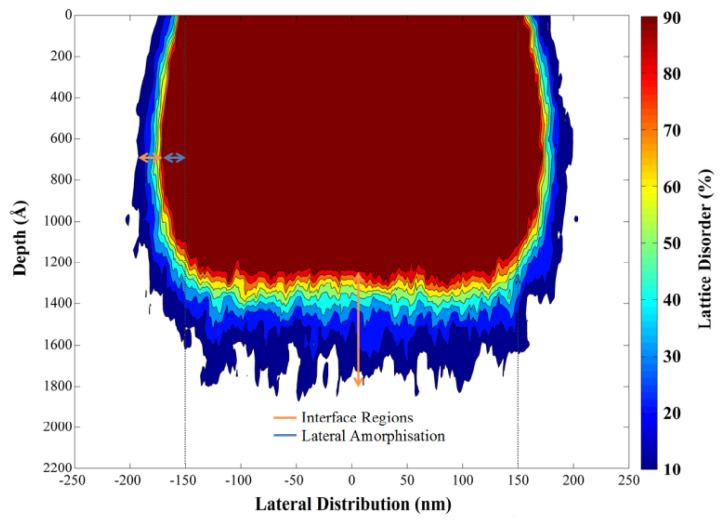
Lattice damage profile at the cross-section of waveguide [45,46,47,48].

**Figure 2 micromachines-13-00291-f002:**
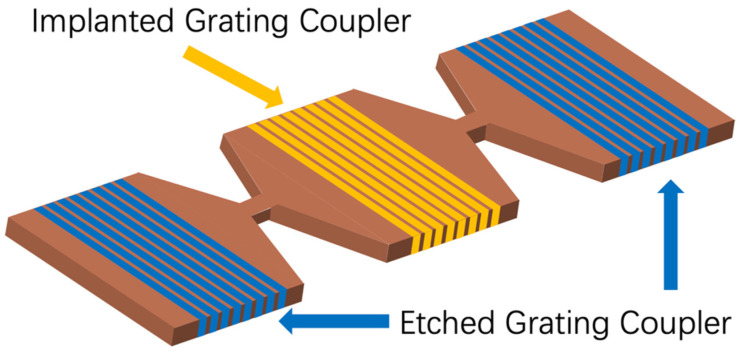
Structure of implanted grating coupler in photonic circuits.

**Figure 3 micromachines-13-00291-f003:**
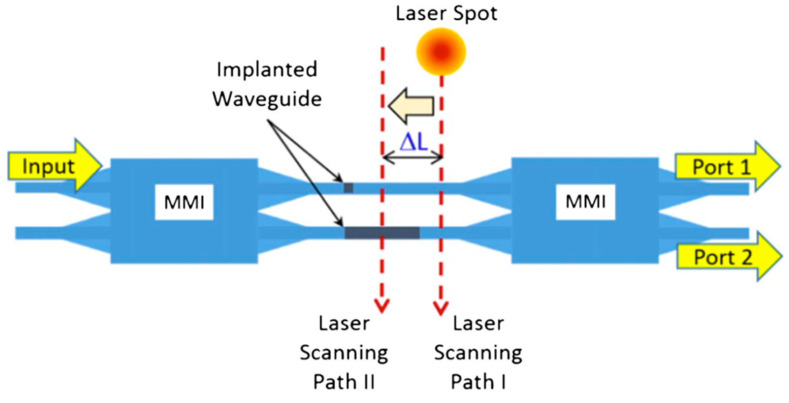
Structure of implanted MZI and scanning route of laser spot [34].

**Figure 4 micromachines-13-00291-f004:**
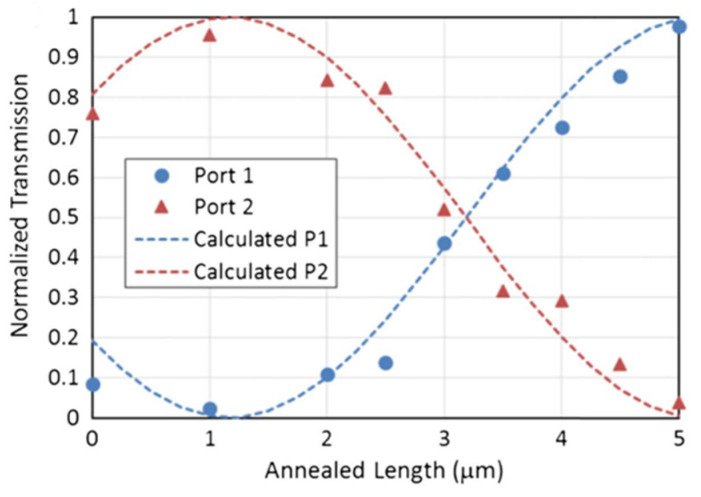
Normalised transmission of implanted MZI during the annealing process [34].

**Figure 5 micromachines-13-00291-f005:**
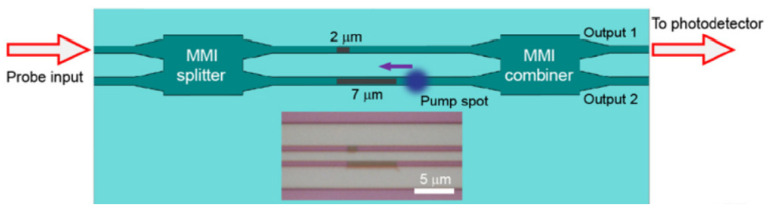
Schematic of real-time pulse laser annealing for implanted MZI [37].

**Figure 6 micromachines-13-00291-f006:**
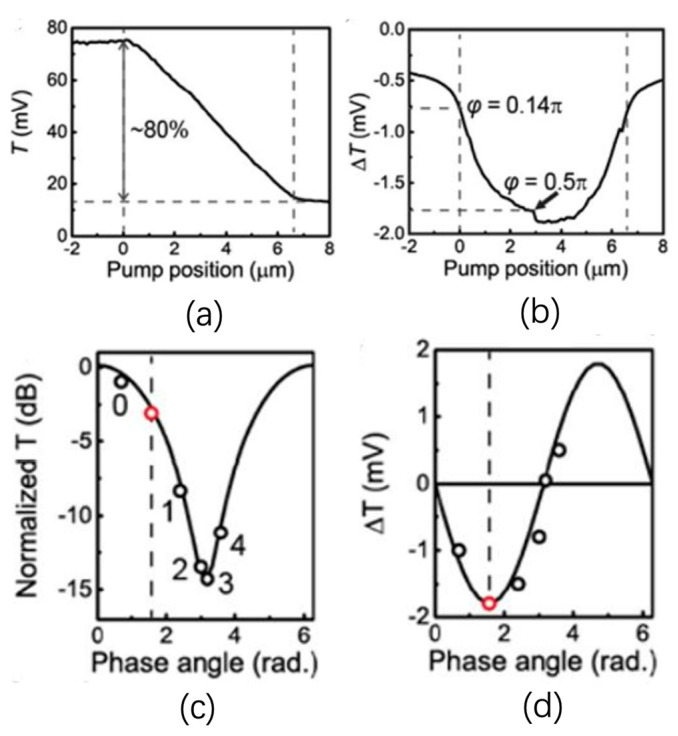
(**a**) Measured transmission (T) signal shifts as the function of the pump spot position during the scanning along the whole 7 μm implanted section; (**b**) measured ΔT signal dependent on the pump spot position. Measured values of (**c**) T and (**d**) ΔT for four subsequent annealing cycles, labelled 1–4 in (**d**), with red dot corresponding to a balanced working point as determined from minimum in ΔT of (**c**) [37].

**Figure 7 micromachines-13-00291-f007:**
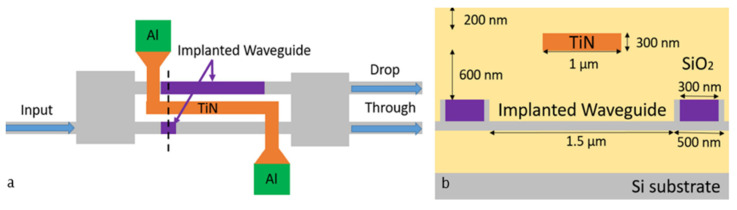
Top view (**a**) and cross-section (**b**) of impanated MZI with the integrated TiN heater on top.

**Figure 8 micromachines-13-00291-f008:**
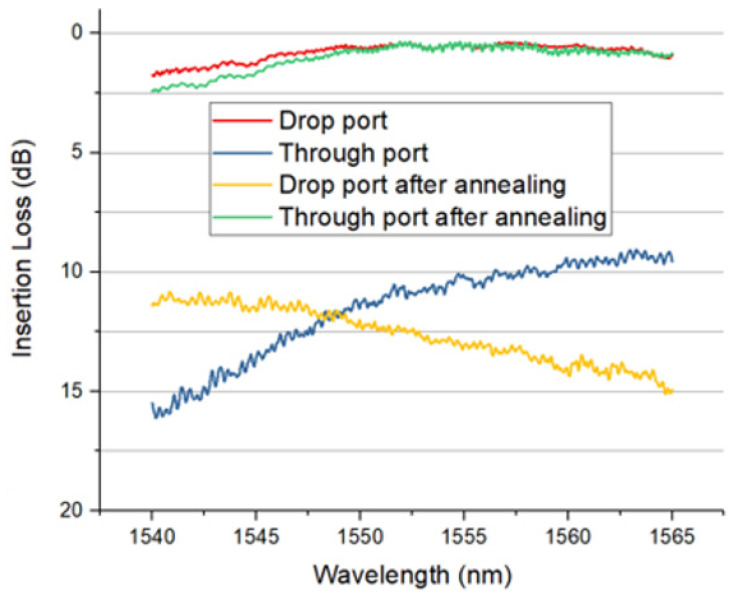
Measured transmission of implanted MZI before and after electrical annealing [19].

**Figure 9 micromachines-13-00291-f009:**
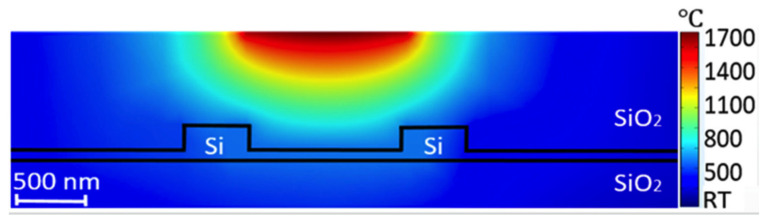
Temperature distribution of implanted MZI at 7 V [19].

**Figure 10 micromachines-13-00291-f010:**
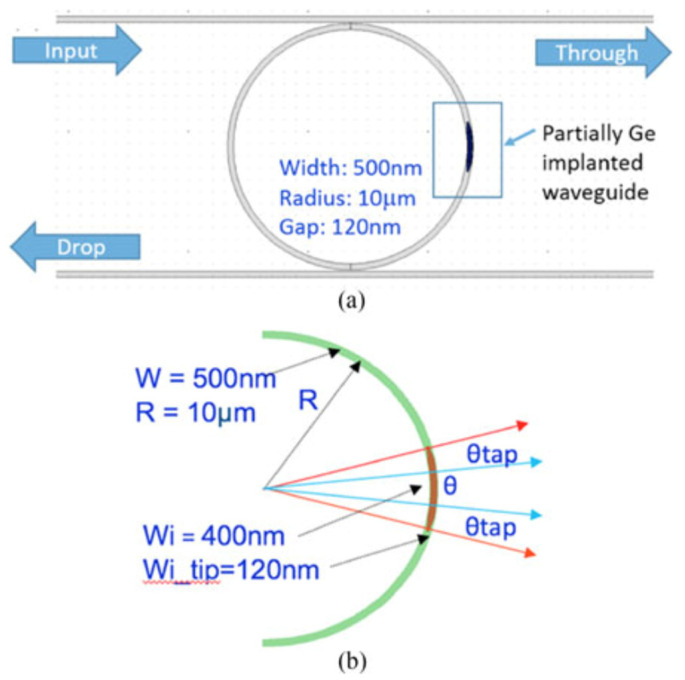
Structure of implanted ring resonator: (**a**) top view of the implanted ring resonator. (**b**) schematic of implanted part in rings [35].

**Figure 11 micromachines-13-00291-f011:**
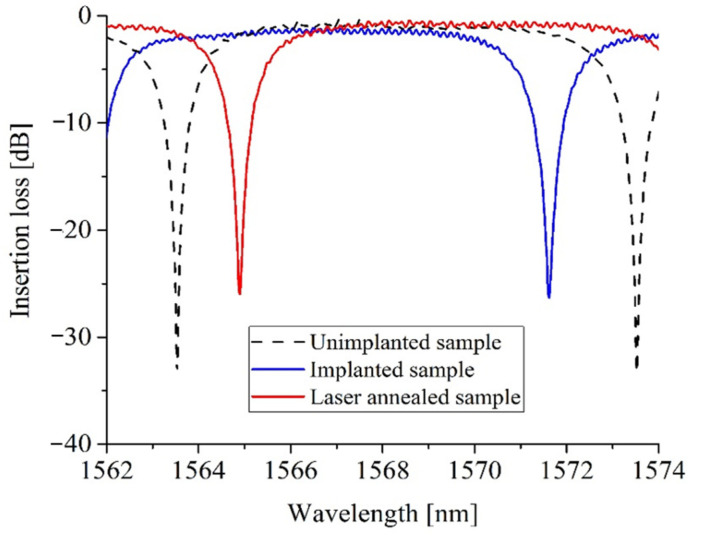
Transmission spectra of implanted ring resonator (with θ = 18°) before and after annealing [35].

**Figure 12 micromachines-13-00291-f012:**
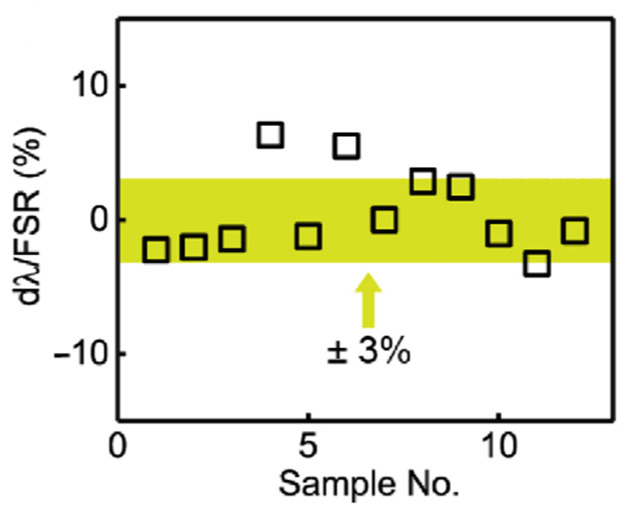
Variation between actual, trimmed resonances and target wavelength 1550 nm normalised to the FSR [37].

**Figure 13 micromachines-13-00291-f013:**
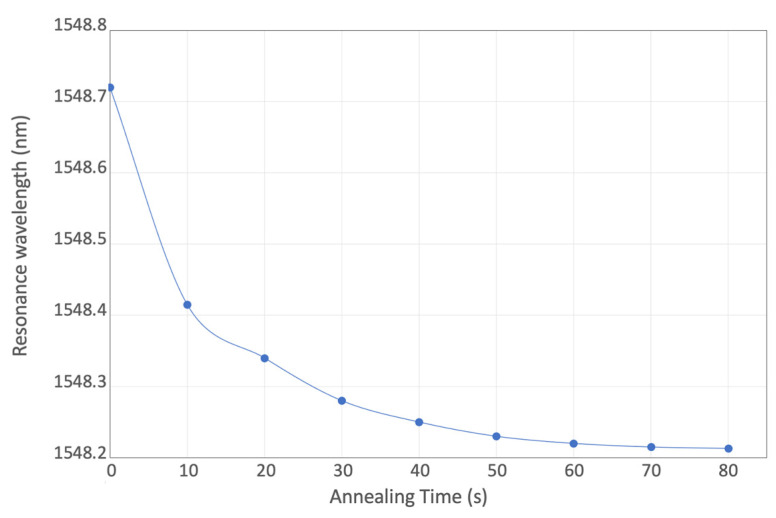
Resonant wavelength shift under 43 mW applied electrical power.

**Figure 14 micromachines-13-00291-f014:**
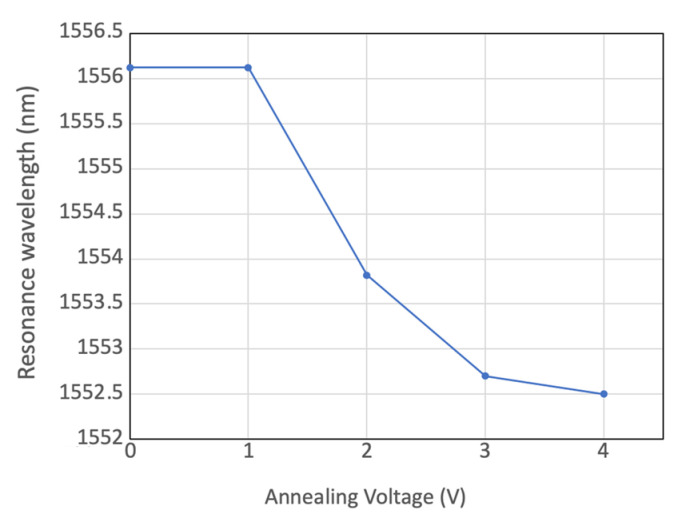
Resonant wavelength shift under applied electrical power of 59 mW, 120 mW, and 160 mW.

**Figure 15 micromachines-13-00291-f015:**
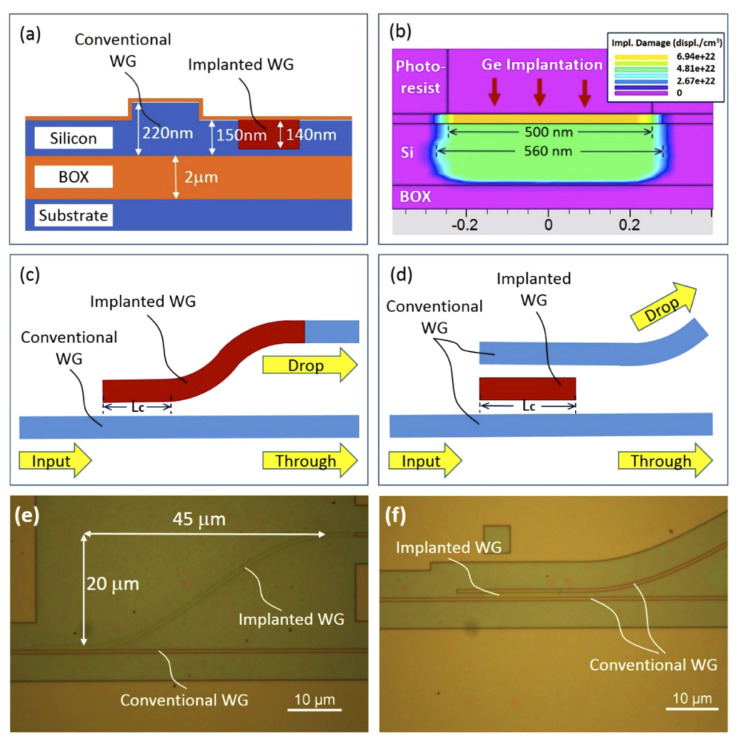
(**a**) Cross-section of implanted waveguide and conventional rib waveguide; (**b**) simulation result for the Ge implanted waveguide using the Silvaco software; (**c**) structure of single-stage DC; (**d**) structure of two-stage DC. Optical microscope images of (**e**) a fabricated single-stage DC and (**f**) a two-stage DC on an SOI wafer [38].

**Figure 16 micromachines-13-00291-f016:**
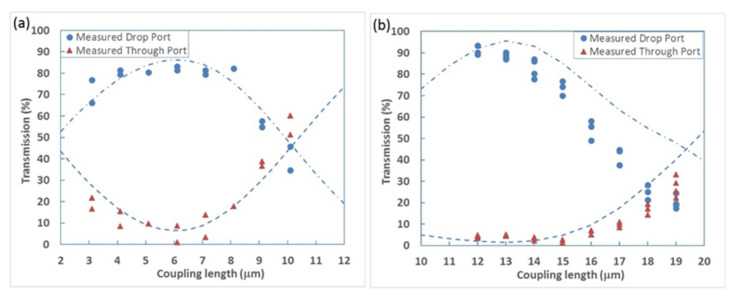
Measured transmission of erasable directional couplers: (**a**) single-stage DCs; (**b**) two-stage DCs [38].

**Figure 17 micromachines-13-00291-f017:**
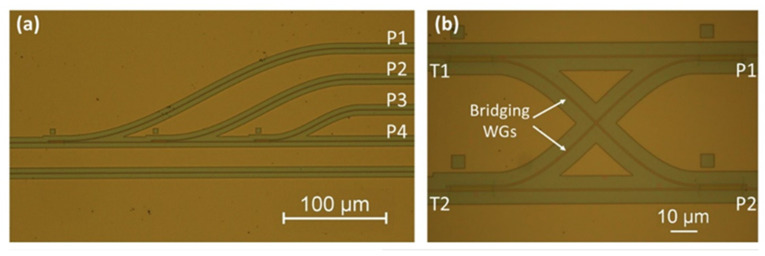
Microscopic image of switching circuits: (**a**) 1 × 4 switch, (**b**) 2 × 2 switch [38].

**Figure 18 micromachines-13-00291-f018:**
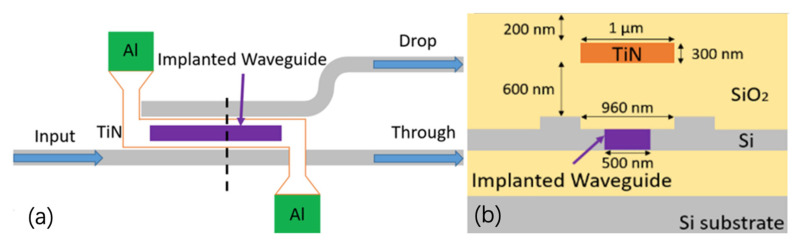
Top view (**a**) and cross-section (**b**) of the ion-implanted two-stage DC [19].

**Figure 19 micromachines-13-00291-f019:**
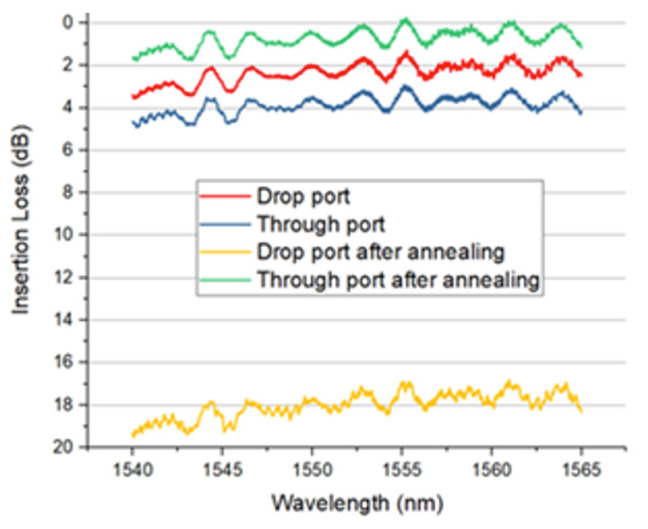
Optical transmission of DC before and after annealing with 19 µm long TiN filament [19].

**Table 1 micromachines-13-00291-t001:** Comparison of current trimming of ring resonators.

	Ref. [29]	Ref. [30]	Refs. [74,75]	Ref. [36]
Methods	E-beam	UV light	Sb_2_Se_3_	Ion implantation and annealing
CMOS compatibility of materials	Yes	No	No	Yes
Throughput (minutes/device)	Low	Low	Low	High
	(>17)	(>10)	(<1)	(<1)
Effective index change	0.06	<0.1	0.017	0.19

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
