# Peer review of "Ge Ion Implanted Photonic Devices and Annealing for Emerging Applications"

_micromachines, 2022, doi:10.3390/mi13020291_

Round 1

Reviewer 1 Report

The manuscript is on implementation of ion implantation in improving the performance of the photonic devices. Above mentioned technique has been employed in various photonic devices like, erasable grating coupler, erasable directional couplers and tuneable MZI and ring resonators. The work looks good and may be of interest to the scientific community working in this area. However, the language needs some improvement. Moreover,
the details on the gist of work, i.e., ion implantation is missing. Authors are encouraged to revise the work with inclusion of below given comments before acceptance:    

1. Line 66 -- Ion implantation is a comparatively sophisticated facility and bears high cost and is not easily available. Please justify in what respect, ion implantation is easy to implement?

2. Line 73-- check and rewrite the sentence.

3. Line 93-- The defects present in the material after ion implantation cause dopants deactivation. Another fact is that generated defects won't anneal 100 percent after annealing. So, please elaborate how the concentration of Si waveguide won't change? Some references may be cited.

4. Line 97-- Please include reference for used software.

5. Line 200-- Please check the sentence and rewrite.

6. Line 205 -- Please include how the selection of the power of laser and implanted region were made?

7. The main stand of the work is ion implantation, however no details on the ions dose, their energy and how their selection was made are not provided. The main advantage of ion implantation, i.e., CONTROLLED modification/introduction of impurities in the various host materials have been widely reported. For example; following works may be referred:

Vibhor Kumar, A.S. Maan, and Jamil Akhtar “Selective SHI irradiation for mesa type edge termination in
semiconductor planar junction” IOP publishing Journal of physics: Conference Series 423, 012057 (2013)  https://doi.org/10.1088/1742-6596/423/1/012057

Vibhor Kumar, A.S.Maan, Jamil Akhtar, "Defect levels in high energy heavy ion implanted 4H-SiC" Materials Letters
308, 131150, (2022). https://doi.org/10.1016/j.matlet.2021.131150

8. Authors are encouraged to include the physics behind how ion implantation improved their device performance in the discussion
section. 

Reviewer 2 Report

This review is undoubtedly useful and interesting, but in this form it cannot be recommended for publication yet and some points need to be clarified more clearly.

  1. In the abstract it is written “silicon induced lattice defects”. Since this manuscript is a "review", an additional paragraph about “silicon induced lattice defects” is simply necessary.
  2. more information needed about current status of photonic materials for photonic devices.
  3. in both of the above, the presence of tables significantly enhance the visibility of the article.
  4. Line 56 - 60. What actually is the difference in what happens when samples are irradiated with light or electrons? We  know, for example,  that in the case of alumina, light leads only to a change in the charge states of impurities and defects, while particle irradiation leads to the formation of new vacancies ( see recent report, introduction and references therein: A. Lushchik, Scientific Reports volume 11, Article number: 20909 (2021) https://www.nature.com/articles/s41598-021-00336-0
  5. Line 83. Note that lattice disorder, as has been shown for several different materials, is highly dependent on the type and fluence (dose) of the irradiation. (Kotomin et al, Phys. Chem. A 2018, 122, 1, 28–32 )
  6. Line 89. “The bonded crystals …. “ -  this sentence is ambiguous.
  7. Line 103. Give more information and discussion about the effect of temperature on radiation damage of Si.

Round 2

Reviewer 2 Report

The authors have significantly improved the original version of this article, taking into account all the recommendations of the reviewers. The article can be recommended for publication